# Lung Cancer Screening in Patients with COPD—A Case Report

**DOI:** 10.3390/medicina55070364

**Published:** 2019-07-11

**Authors:** Roxana Amirahmadi, Avnee J. Kumar, Mark Cowan, Janaki Deepak

**Affiliations:** 1Department of Medicine, University of Maryland School of Medicine, Baltimore, MD 21201, USA; 2Division of Pulmonary and Critical Care Medicine, Department of Medicine, University of Maryland School of Medicine, Baltimore, MD 21201, USA; 3Division of Pulmonary and Critical Care Medicine, Baltimore VA Medical Health Center, Baltimore, MD 21201, USA

**Keywords:** lung cancer screening, COPD, lung nodule, low dose computed tomography

## Abstract

We present two cases demonstrating the nuances that must be considered when determining if a patient could benefit from low dose computed tomography (LDCT) lung cancer screening. Our case report discusses the available literature, where it exists, on lung cancer screening with special attention to the impact of chronic obstructive pulmonary disease (COPD), and poor functional status. Patients with COPD and concurrent smoking history are at higher risk of lung cancer and may therefore benefit from lung cancer screening. However, this population is at increased risk for complications related to biopsies and lobar resections. Appropriate interventions other than surgical resection exist for COPD patients with poor pulmonary reserve. Risks and benefits of lung cancer screening are unique to each patient and require shared decision-making.

## 1. Introduction

Lung cancer is the second most common malignancy worldwide, resulting in one-quarter of all cancer-related deaths [1]. Timely lung cancer screening and early detection has been shown to favorably impact outcomes. In particular, the National Lung Cancer Screening Trial (NLST) showed that high-risk patients benefit from low dose computed tomography (LDCT) annual screening, with a 20% decrease in mortality [2]. However, several post-hoc studies following the NLST suggested that not all subpopulations of patients benefit to the same extent [3,4]. There is limited evidence elucidating how alternate risk factors for lung cancer, such as occupational exposures or chronic obstructive pulmonary disease (COPD), affect a patient’s probability for experiencing risk or benefit from lung cancer screening [5]. It has been shown that patients with radiographic or spirometric evidence of COPD, including emphysema, are more likely to develop lung cancer than their counterparts without evidence of COPD [6,7].

The decision to screen a patient with COPD is complex and worth further study for several reasons. First, COPD is an independent risk factor for the development of lung cancer. Second, cigarette-smoking is a strong risk factor for both COPD and lung cancer, as the mechanisms linking COPD and lung cancer are numerous and complex. Here, we present a case report of the literature, outlining factors to consider when determining if a patient with COPD should be screened for lung cancer.

## 2. Case Vignettes

### 2.1. Vignette 1

A 66-year-old Caucasian gentleman with gastroesophageal reflux disease, hyperlipidemia, 40-pack-year smoking history, and COPD presented for LDCT screening. At baseline, he could walk ten blocks on level ground and climb up two flights of stairs. Home medications included triple therapy with a long-acting muscarinic antagonist, a long-acting beta agonist, and an inhaled corticosteroid. Pulmonary function tests (PFTs) showed an FEV_1_ of 1.08 L (36%), FVC of 2.44 L (56%), FEV_1_/FVC ratio of 44, TLC of 8.63 L (136%), and DLCO of 15.6 mL/min/mmHg (92%).

Results of the LDCT showed a new spiculated 1.7 cm × 1.7 cm nodule in the left lower lobe. The patient underwent subsequent positron emission tomography/computed tomography (PET/CT), which showed an isolated hypermetabolic lesion in the left lower lobe with a standardized uptake value (SUV) of 6.1 and uptake in the mediastinal lymph nodes. He was referred for endobronchial ultrasound guided transbronchial needle aspiration (EBUS-TBNA) of lymph node stations 11 L, 7, 4L, 4R, 11 R, in addition to biopsy of the left lower lobe nodule. Pathology revealed squamous cell carcinoma in the left lower lobe with negative lymph node staging. Given the size of the lesion, the negative lymph node survey, and PET/CT with an isolated lesion, his cancer was T1bN0M0 or stage IA2 squamous cell cancer of the lung.

Despite his severe obstructive defect with hyperinflation, air trapping, and reduced FEV_1_, he had excellent performance status. The patient performed adequately on cardiopulmonary exercise testing. The patient exercised for 9-min, and his aerobic capacity was mildly reduced with a peak VO_2_ of 22 mL/Kg/min. Maximum work obtained was 95 Watts. 6-min walking distance measured 650 feet with oxygen-operative quantitative V/Q scan showing that the left lower lobe had 10.6% perfusion with similar ventilation, and he was deemed to be a good surgical candidate. The patient was able to stop smoking a month later. Two months after smoking cessation, he underwent robotic-assisted left lower lobectomy. The patient deferred treatment for two months due to personal issues. At the time of resection, the size of the tumor in the left lower lobe was 2.6 cm with a negative lymph node survey. Post-lobectomy, his pathological staging was T1cN0M0 or stage IA3 squamous cell cancer of the left lower lobe. He has been followed in surgery clinic with chest CT every three months for the first two years, then annually. The patient is still alive 3 years after surgery. The follow-up surveillance CT scans show no recurrence so far.

### 2.2. Vignette 2

A 69-year-old Caucasian gentleman with coronary artery disease status post 6-vessel coronary artery bypass grafting, obstructive sleep apnea, a 60-pack-year smoking history, and COPD on 2 L/hour of oxygen presented for LDCT screening. At baseline, he could perform all activities of daily living, however, became short of breath when climbing two flights of stairs. Pulmonary function tests (PFTs) performed just before our evaluation showed an FEV_1_ of 1.83 L (75%), FVC of 2.85 L (85%), FEV_1_/FVC of 64, TLC of 6.02 L (128%), and DLCO of 12.7 mL/min/mmHg (94%), indicating mild obstruction with hyperinflation and preserved gas transfer.

Results of the LDCT showed a new spiculated 1.5 cm × 1.0 cm solid nodule in the right upper lobe, and severe pan-lobar emphysema with minimal intervening normal lung parenchyma. The patient underwent subsequent PET/CT, which showed isolated avidity in the right upper lobe nodule with an SUV of 5, and no other metabolically avid lesions. He was a poor candidate for bronchoscopic evaluation given his oxygen requirement and intermittent desaturations. Therefore, he was referred for transthoracic needle aspiration of the right upper lobe nodule. Pathology revealed invasive moderately differentiated adenocarcinoma. His echocardiogram showed LVEF 55%, and his left heart catheterization showed severe native triple disease and restenosis of the vein graft to LAD requiring drug-eluting stent placement. Given his poor functional status and his comorbid conditions, he was referred for stereotactic body radiation therapy (SBRT). However, he was unable to withstand the mapping CTs needed for this, as well as MRI, due to severe rib fractures sustained secondary to sleepwalking and multiple falls due to non-compliance with CPAP for his OSA. In addition, he also has severe claustrophobia. The patient was too unsteady on his feet to perform exercise testing. Hence, the patient underwent radiofrequency ablation of the right upper lobe nodule. Post ablation, his staging was T1N0M0 or stage IA2 adenocarcinoma of the lung. The patient is 6 years progression-free without recurrence, and he is still alive.

## 3. Discussion

Smokers with COPD have up to a 6-fold increased risk of lung cancer compared to smokers with normal lung function. In a Danish lung cancer screening trial cohort, patients with an obstructive defect on PFTs, evidence of emphysema on chest CT, greater than or equal to 35-pack-year smoking history, and age greater than 70, had two-times greater risk of death related to lung cancer than other subgroups [8]. Additionally, simply having COPD and, in turn, increased stem cell recruitment, accelerated cell turnover, and relatively elevated levels of local and systemic inflammation independently predispose the patient to a higher risk of lung cancer [6,9]. Likely as a result, patients with COPD are shown to benefit more from lung cancer screening than comparable patients without COPD, as their tumors tend to be caught at earlier stages and are less likely to be falsely positive [10]. While patients with COPD appear to have much to gain from lung cancer screening, these patients are also at a higher risk of complications from surgical interventions and are less tolerant of chemotherapy or radiation treatments [11]. This increases decision-making complexity when weighing the risks and benefits of LDCT lung cancer screening in patients with COPD.

Here we present two cases of patients with COPD and considerable smoking history who underwent lung cancer screening at our institution. Our first case highlights a patient with severe COPD in which lung cancer screening detected an early stage lesion. Despite a severe obstructive defect, this patient was successfully treated with surgical resection. Importantly, the patient had relatively good functional capacity, which likely contributed greatly to his ability to tolerate surgery. This brings up the importance of evaluation of functional capacity prior to deciding to screen the patient for lung cancer, despite the FEV_1_. Furthermore, obtaining quantitative ventilation-perfusion (V/Q) scans before lobar resections is likely a valuable tool to determine the postoperative FEV_1_ and operative risk after finding a nodule. This patient may have presented with a later stage lung cancer and potentially worse outcomes without LDCT. This case illustrates that even COPD patients with significant obstructive patterns on PFTs should be considered for lung cancer screening if they have good functional status.

Our second patient’s options were limited by his frequent desaturations. Although he did not have severe COPD on PFTs, his supplemental oxygen requirement and functional status limited complex pulmonary procedures. This highlights the importance of therapies other than surgical resection for early-stage lung cancer. The most commonly used non-surgical modality of treatment for early-stage lung cancer is stereotactic body radiation therapy (SBRT). SBRT is a noninvasive cancer treatment in which numerous small, highly focused, and acute rate radiation beams are used to deliver potent doses in 1 to 5 treatments to tumor targets and extracranial sites. However, ablative techniques, like radiofrequency ablation (RFA), cryoablation, and microwave ablation, are also viable alternatives [12,13]. A large study cohort of stage IA and IB non-small cell lung cancer treated with primary radiofrequency ablation or SBRT showed higher risk for unplanned readmission within 30 days in the RFA cohort. However, there was no difference in the overall survival between the 2 groups. Furthermore, prior systematic reviews of 44 studies comparing RFA and SBRT have similarly shown no significant difference in the overall survival. These less invasive and better tolerated alternatives to chemotherapy and resection expand the population of patients with severe COPD who may benefit from LDCT, as our patient did. However, when discussing treatment options to COPD patients who are poor surgical candidates, it is important to keep in mind that RFA is only recommended for tumors that are ≤3 cm in size, as tumors greater than this size are associated with less progression-free survival [14]. We need to further study and better quantify the survival benefits of RFA for our patients who are poor surgical candidates, so we can use this body of data in shared decision making.

A consideration for all patients prior to screening is the risk of false positives and overdiagnosis of indolent tumors, along with the possibility of subsequent exposure to unnecessary radiation, procedures, and anxiety [5]. Several studies following the NLST raised concerns that routine LDCT screening led to overdiagnosis of potentially indolent tumors. There were 120 more lung cancer nodules found on LDCT compared to the CXR arm in the NLST, and while a recent systematic review concluded that further investigation into the rate of overdiagnosis in the practice of routine LDCT screening [5], another study by Patz et al. reported an estimated rate of overdiagnosis could be as high as about 1 out of 5 of all lung cancer tumors detected in NLST [3]. Another study published in Annals of ATS by Thalanayar et al. followed 93 patients in the Pittsburgh Lung Screening Study (PLuSS) cohort, and identified that 18.5% of lung cancers identified were indolent tumors, which the study defined as stage I nodule on LDCT with a volumetric doubling time >400 days, and standardized uptake value max (SUVmax) ≤1 on PET scan. The median doubling time of these indolent tumors was 752 days, compared to 284.5 days in the non-indolent stage 1 cancers. This study also found that the patient characteristics in the cohort with indolent versus non-indolent tumors were not significantly different (including presence of emphysema or average FEV_1_). Additionally, it is important to keep in mind that the growth of a tumor may not be constant over time, and prior patterns of growth may not be indicative of future rates of growth [15]. Therefore, more research needs to be done over long periods of time to see if study the incidence of indolent tumors in the COPD patient is more or less likely to have indolent cancers found on LDCT screenings, and to better characterize the natural progression of these indolent cancers over the long-term in COPD patients.

However, some literature suggests that COPD patients may benefit more than other patients who undergo lung cancer screening. A study on NLST patients compared the benefit of lung cancer screening specifically in patients with COPD to patients without COPD. Patients with spirometry suggestive of COPD were two times more likely to get lung cancer over a 6-year period. When lung cancer was discovered on LDCT screening, there was a higher ratio of stage 1–2 to stage 3–4 lung cancers found compared to patients who underwent chest radiography. However, this higher proportion of early-stage lung cancers discovered on LDCT did not come at the cost of higher false positive rate in the COPD arm [10]. This suggests that low-dose CT is not only more effective in discovering earlier stages of lung cancer in COPD patients compared to patients without COPD, but also does so with a lower false positive rate. Despite this, while neither of our patients had adverse events, statistically speaking, patients with COPD are more likely to suffer from complications related to lobar resections and fine needle biopsies [11]. Therefore, the consequences of unnecessary biopsies and lobe resections should not be neglected when considering the downstream risks of lung cancer screening in a COPD patient [13].

The complexity of screening in patients with COPD calls into question if there is an established thought process that physicians should use when considering screening in a COPD patient. Analysis of the Pamplona International Early Lung Cancer Detection Program (P-IELCAP), and the PLuSS by Torres et al. yielded a scoring tool to stratify patients with COPD undergoing lung cancer screening. The COPD lung cancer screening score, or COPD-LUCSS, stratifies patients into ‘high risk’ and ‘low risk’ for lung cancer. The score ranges from 0 to 10 points, and includes age greater than 60, body mass index less than 25 kg/m2, pack-years history, and emphysema presence [16,17]. Our first patient would be categorized as high risk with 7 points given his age and radiologic presence of emphysema. Our second patient would be categorized as high risk with 9 points given his pack-year history, age, and radiologic presence of emphysema. Use of this score in shared decision making is another factor to play into the decision to screen for lung cancer. For example, if a patient is low risk using this tool and has poor functional capacity, LDCT may not be warranted. To add an additional layer of complexity, analysis of the P-IELCAP and PLuSS participants showed that NLST screening guidelines missed up to 39% of lung cancers. Broadening screening to patients that met either NLST criteria, or had emphysema detected 88% of lung cancer in P-IELCAP and 95% in the PLuSS. Overall, the composite fitness of a patient and the other comorbidities that might further limit a patient’s lifespan may be the most valuable factors when determining if a COPD patient should undergo LDCT. Besides the COPD-LUCSS-DLCO score, which replaces CT-determined emphysema with DLCO (which was presumed to be a more clinically attainable data point than CT scan) as a surrogate for presence of emphysema [18], there is no significant literature validating other lung cancer screening scores for patients with COPD.

Assessing a patient’s fitness for thoracic surgery can help clinicians determine if a patient should undergo LDCT lung cancer screening. Clinicians may use the 6-min walking test, echocardiography, or even stress testing to assess a patient’s fitness prior to surgery. One available score is the Thorascore, which uses nine parameters to predict inpatient mortality after thoracic surgery. Thangakunam et al. assessed the effectiveness of Thorascore in evaluating fitness of lung cancer patients prior to thoracic surgery compared to cardiopulmonary testing. They found that there was no correlation between a patient’s Thorascore and their lung function parameters, duration of hospital stay after surgery, or peak VO2 [19]. The European Society of Thoracic Surgery and the European Respiratory Society taskforce made several recommendations using a composite of the body of literature available at that time (in 2009) on how to risk stratify lung cancer patients for surgery and radio/chemotherapy. This taskforce offered the Grade B recommendation that routine pre-operative testing of DLCO should be done before lung resection surgery, even if the patient never demonstrated evidence of spirometric abnormalities, and a DLCO of <30% predicted is high risk for compromised pulmonary reserve [20]. However, this taskforce did not find that pulmonary function testing was as useful when risk stratifying lung cancer patients for radiotherapy modalities. More studies need to be done to develop decision tools and risk assessment strategies for COPD patients prior to initiating lung cancer screenings.

Office visits dedicated to discussing the risks and benefits of lung cancer screening have proven effective in enhancing patient knowledge of lung cancer screening criteria [21]. A prospective study from Temple University found that devoting a single clinic visit to evaluate underserved patients’ willingness to undergo lung cancer treatments if lung cancer is found was clinically feasible. More importantly, the study showed that protected clinic time to discuss the patient’s risks/benefits of screening did not significantly discourage patients from undergoing screening [22]. However, these results might not be generalizable to COPD patients, as they often have multiple comorbidities that might make finding the ‘correct path’ more challenging. Nonetheless, devoted time for discussion and shared decision making will likely result in a better chance of arriving at a screening strategy best for each patient.

## 4. Conclusions

Low dose CT scan screening is crucial in patients with COPD given their elevated risk of lung cancer. However, their ability to undergo subsequent procedures is limited by functional status. Careful evaluation of comorbid conditions, provider knowledge of potential treatment options, and shared decision making prior to screening is essential. The question to screen or not screen is complex and entails looking at the individual patient. Performance status, PFTs, imaging, and comorbidities that will affect candidacy for further testing and a broad range of surgical and non-surgical treatment modalities must be carefully considered. Shared decision making is a vital strategy that physicians can use early on to help the patient with COPD to navigate their goals and enhance their knowledge.

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
