# Peer review of "Lung Cancer Screening in Patients with COPD—A Case Report"

_medicina, 2019, doi:10.3390/medicina55070364_

Round 1
Reviewer 1 Report
I kindly reccomend to: - increase the discussion section with more bibliographic indexes - a short information about survival rate for both cases only if it is availableAuthor Response
COMMENT #1: increase the discussion section with more bibliographic indexes
RESPONSE #1: Thank you for your feedback. We have included some more bibliographic sources.
COMMENT #2: a short information about survival rate for both cases only if it is available
RESPONSE #2: We appreciate your suggestion. The patient in case 1 is still alive 3 years after surgery. The follow up surveillance CT scans show no recurrence so far. Patient in case 2 has 6 years of progression free survival.
Reviewer 2 Report
I congratulate for this excellent and interesting work. I have some minor comments:
Gammatic: in Line 121 you write "IBM", is here the M not unnecessary?
Some theoretical comments, questions for wchich the answers are missing from the paper:
For Case 1: I am interested in the six minute walking distance, and about any form of pulmonary rehabilitation. What was the reason for waiting 2 month for the operation? As you correctly discribed, the tumor increased in this time, fortunately not escaleted in the lymph nodes. I am interested in the Progression Free Survival also.
For Case 2: For this case i miss some important results: ECHO or Cariac MRI result, six-minute walking test, blood gas results in resting positon, Ergometric, or ergospirometric result. In the therapy not only the Oxygen supplementation, but also the other COPD and cardiac therapy could be intresting. What was the reason, that the patient was unable to withstand the mapping CTs?
I like the discussion part of your work. It is very important to speak about risks of screening and lung cancer diagnostic interventions in this patients population, and also about higher risk for lung cancer, and lower false positive rate in LDCT-s.
I would read more about all the mentioned lung cancer screening scores, and also about scores and interventions recommended to evaluate "the composite fitness of a patient".
Author Response
COMMENT #1: Gammatic: in Line 121 you write "IBM", is here the M not unnecessary?
RESPONSE #1: Thank you for your suggestion. You are correct; we changed “IBM” to IB.
COMMENT #2: For Case 1: I am interested in the six minute walking distance, and about any form of pulmonary rehabilitation. What was the reason for waiting 2 month for the operation? As you correctly discribed, the tumor increased in this time, fortunately not escaleted in the lymph nodes. I am interested in the Progression Free Survival also.
RESPONSE #2: Thank you for your feedback. Adequate CPET. The patient exercised for 9 minutes. Aerobic Capacity is mildly reduced with a peak VO2 of 22 ml/Kg/min. Maximum work obtained was 95 Watts. VQ scan showed 10% left lower lobe perfusion. 6MWD 650 feet with oxygen. Patient still alive for 1 year after surgery. 6 month and 1 Year Ct no recurrence so far. Reason for the 2 month wait was patient deferring any treatment for personal issues.
COMMENT #3: For Case 2: For this case i miss some important results: ECHO or Cariac MRI result, six-minute walking test, blood gas results in resting positon, Ergometric, or ergospirometric result. In the therapy not only the Oxygen supplementation, but also the other COPD and cardiac therapy could be intresting. What was the reason, that the patient was unable to withstand the mapping CTs?
RESPONSE #3: Thank you for your great points. Reasons for inability to undergo CT mapping were multiple falls leading to severe rib fractures secondary to sleepwalking. Patient has OSA non-compliant with CPAP and severe claustrophobia, so he refused an MRI. Te also was too unsteady to perform exercise testing. ECHO showed LVEF 55%, left heart catheterization showed severe native triple disease and restenosis of the vein graft to LAD requiring drug-eluting stent placement. Patient has 6 year progression free survival and is still alive.
COMMENT #4: I like the discussion part of your work. It is very important to speak about risks of screening and lung cancer diagnostic interventions in this patients population, and also about higher risk for lung cancer, and lower false positive rate in LDCT-s.
RESPONSE #4: Thank you very much. We made sure to include a well-rounded discussion of how to assess if our COPD patients can handle diagnostic and therapeutic interventions that they may need to undergo before making the decision that LDCT lung cancer screening is appropriate for them. We have included a discussion of the body of research showing how COPD patients are actually at a lower risk of false positives and higher risk for lung cancer.
COMMENT #5: I would read more about all the mentioned lung cancer screening scores, and also about scores and interventions recommended to evaluate "the composite fitness of a patient".
RESPONSE #5: Thank you for your suggestions. Besides the scoring tools we included in our paper previously and the COPD-LUCSS-DLCO score, which replaces the use of CT-determined emphysema with DLCO as a surrogate for presence of emphysema, we did not find any other lung cancer screening scores for patients with COPD.
We included further details on the assessment of patient fitness prior to thoracic surgery as well as radiotherapy. We found the Thorascore, a multivariate scoring system using nine parameters to predict inpatient mortality after thoracic surgery. Thangakunam et al assessed the Thorascore’s effectiveness in evaluating fitness of lung cancer patients prior to thoracic surgery compared to cardiopulmonary testing. They found that there was no correlation between a patient’s Thorascore and their lung function parameters, duration of hospital stay after surgery, or peak VO2.
The European Society of Thoracic Surgery and the European Respiratory Society taskforce made several recommendations using a composite of the body of literature available at that time (in 2009) on how to risk stratify lung cancer patients for surgery and radio/chemotherapy. This taskforce recommended routine testing of DLCO pre-operatively before lung resection surgery even if the patient never demonstrated evidence of spirometric abnormalities. DLCO of < 30% predicted is considered high risk for compromised pulmonary reserve. The taskforce gave this recommendation a level of evidence of 2++ and presented it as a Grade B recommendation. However, this taskforce did not find that pulmonary function testing offered the same predictive value when risk stratifying lung cancer patients for radiotherapy modalities. More studies need to be done to develop decision tools and risk assessment strategies for COPD patients who might undergo lung cancer screening.
Reviewer 3 Report
This an an excellent review of an important and evolving topic in the care of advanced COPD patients. The authors do an outstanding job of reviewing this topic and the added importance of lung cancer screening in appropriate candidates who also have a diagnosis of COPD. It is a well researched and clinically sound review.
I have some minor comments I hope the authors can address.
The current guidelines and landmark studies recommend discontinuing screening in patients who are no longer capable/unwilling of having curative lung surgery. Case 2 is a very realistic situation that often places the well intentioned clinician in a position to explore alternatives for lung cancer therapy. However it is important to note and stress to the readers that whether such therapies (radiation or RFA) are associated with the same or any survival benefit in screened patients is unclear at this time and should be studied in the future.
Dove tailing with the above comments, I would recommend the authors make a comment about overdiagnosis in the same way they comment on false positive screen. There are likely indolent cancers found with screening that will not change life expectency especially in high risk patients and empiric radiation or RFA which are being recommended more and more for these patients may be inherently harmful given the complications of these treatments to high risk patients. I am unaware of a decision tool of deciding which tumors to treat and not treat (Is their a recommended size? other characteristics? etc)
On page 3 in the third paragraph (129-131), I think the wording of this sentence is misleading. NLST showed that of 1000 patients screening with LDCT, 18 died of lung cancer while 21 not screening died of lung cancer. That means 3 patients of the 1000 patients screening will be saved/alive with screening. Saying that 3 of 1000 had their lung cancer detected is not the same thing.
Author Response
COMMENT #1: The current guidelines and landmark studies recommend discontinuing screening in patients who are no longer capable/unwilling of having curative lung surgery. Case 2 is a very realistic situation that often places the well intentioned clinician in a position to explore alternatives for lung cancer therapy. However it is important to note and stress to the readers that whether such therapies (radiation or RFA) are associated with the same or any survival benefit in screened patients is unclear at this time and should be studied in the future.
RESPONSE #1: This is a great point. Thank you for your feedback.
We found that The European Society of Thoracic Surgery and the European Respiratory Society taskforce made several recommendations using a composite of the body of literature available at that time (in 2009) on how to risk stratify lung cancer patients for surgery and radio/chemotherapy. This taskforce offered the Grade B recommendation that routine pre-operative testing of DLCO should be done before lung resection surgery, even if the patient never demonstrated evidence of spirometric abnormalities, and a DLCO of < 30% predicted is high risk for compromised pulmonary reserve. [20] However, this taskforce did not find that pulmonary function testing was as useful when risk stratifying lung cancer patients for radiotherapy modalities. More studies need to be done to develop decision tools and risk assessment strategies for COPD patients prior to initiating lung cancer screenings.We have included a statement that we need further study if patients who undergo RFA have higher rates of survival.
COMMENT #2: Dove tailing with the above comments, I would recommend the authors make a comment about overdiagnosis in the same way they comment on false positive screen. There are likely indolent cancers found with screening that will not change life expectency especially in high risk patients and empiric radiation or RFA which are being recommended more and more for these patients may be inherently harmful given the complications of these treatments to high risk patients. I am unaware of a decision tool of deciding which tumors to treat and not treat (Is their a recommended size? other characteristics? etc)
RESPONSE #2: Thank you for your comments. Recognizing the role of overdiagnosis of indolent tumors that may not limit the life expectancy is an important issue to raise, especially in COPD patients who usually have several lung cancer risk factors. We have included a discussion of the role of indolent tumors given the significant rate of indolent tumors found in a study by Thalanayar et al, which estimated the rate of indolent tumors found in the PLuSS patient cohort and estimated that about 18.5% of the lung cancers identified were indolent tumors, which the study defined as stage I nodule on LDCT with a volumetric doubling time >400 days and standardized uptake value max (SUVmax) ≤1 on PET scan.
The recommended size for RFA is less than 3 cm; beyond this there is increased risk with less progression free survival. We will mention this in our paper and cite relevant sources.
COMMENT #3: On page 3 in the third paragraph (129-131), I think the wording of this sentence is misleading. NLST showed that of 1000 patients screening with LDCT, 18 died of lung cancer while 21 not screening died of lung cancer. That means 3 patients of the 1000 patients screening will be saved/alive with screening. Saying that 3 of 1000 had their lung cancer detected is not the same thing.
RESPONSE #3: Thank you very much for bringing up this point. We have adjusted our discussion of the NLST findings so that it is not misleading.